# Phosphodiesterase 5 Inhibitor Potentiates Epigallocatechin 3-O-Gallate-Induced Apoptotic Cell Death via Activation of the cGMP Signaling Pathway in Caco-2 Cells

Jaehoon Bae [1], Kwanwoo Lee [2], Ji-Sun Park [1], Jinseok Jung [1], Hirofumi Tachibana [2], Yoshinori Fujimura [2], Motofumi Kumazoe [2], Jae Sung Lim [3], Young-Chang Cho [3], Seung-Jae Lee [1,*] and Su-Jin Park [1,*]

[1] Functional Biomaterial Research Center, Korea Research Institute of Bioscience and Biotechnology, 181 Ipsin-gil, Jeongeup-si 56212, Jeonbuk, Republic of Korea

[2] Division of Applied Biological Chemistry, Department of Bioscience and Biotechnology, Faculty of Agriculture, Kyushu University, Fukuoka 812-8581, Japan

[3] College of Pharmacy and Research Institute of Pharmaceutical Sciences, Chonnam National University, 77 Yongbong-ro, Gwangju 61186, Republic of Korea

* Correspondence: seung99@kribb.re.kr (S.-J.L.); sjpark@kribb.re.kr (S.-J.P.); Tel.: +82-(63)-570-5267 (S.-J.L.); +82-(63)-570-5240 (S.-J.P.)

**Abstract:** Epigallocatechin 3-*O*-gallate (EGCG) is a predominant component in green tea with various health benefits. The 67 kDa laminin receptor (67LR) is a nonintegrin cell surface receptor that is overexpressed in various types of cancer; 67LR was identified a cell surface EGCG target that plays a pivotal role in tumor growth, metastasis, and resistance to chemotherapy. However, the plasma concentration of EGCG is limited, and its molecular mechanisms remain unelucidated in colon cancer. In this study, we found that the phosphodiesterase 5 (PDE5) inhibitor, vardenafil (VDN), potentiates EGCG-induced apoptotic cell death in colon cancer cells. The combination of EGCG and VDN induced apoptosis via activation of the endothelial nitric oxide synthase/cyclic guanosine monophosphate/protein kinase Cδ signaling pathway. In conclusion, the PDE5 inhibitor, VDN, may reduce the intracellular PDE5 enzyme activity that potentiates EGCG-induced apoptotic cell death in Caco-2 cells. These results suggest that PDE5 inhibitors can be used to elevate cGMP levels to induce 67LR-mediated, cancer-specific cell death. Therefore, EGCG may be employed as a therapeutic candidate for colon cancer.

**Keywords:** epigallocatechin 3-*O*-gallate; vardenafil; phosphodiesterase 5; cyclic guanosine monophosphate; endothelial nitric oxide synthase



## 1. Introduction

Epigallocatechin 3-*O*-gallate (EGCG) is an effective anticancer polyphenol in green tea [1]. Previously, we identified a cell surface protein, 67 kDa laminin receptor (67LR), for EGCG [2]. Several types of cancer cells overexpress 67LR at cell surfaces [3]. Therefore, the plasma concentration of EGCG is limited to induce a killing effect in cancer cells [4]. In addition, little is known about the molecular mechanisms underlying EGCG-induced effects in colorectal cancer cells. Previous studies focused on the inhibitory effect of EGCG on PDEs [5], and those concentrations are quite high considering the clinical settings. Our scope is focused on the inhibition of PDE5, which may be helpful for enhancing the anticancer effect of EGCG on colon cancer cells. Our results showed that PDE5 knockdown is sufficient to enhance the effect of EGCG in colon cancer. Moreover, we provide a clinical evaluation of its effects.

Phosphodiesterase 5 (PDE5) is a specific negative regulator of cyclic guanosine monophosphate (cGMP) [6]. It is overexpressed in several cancer cells [3]. The inhibition of PDE5 activity to induce intracellular cGMP production might be a useful therapeutic

approach for various diseases and conditions, including erectile dysfunction [7–9]. Nitric oxide (NO) is a gasotransmitter that regulates soluble guanylyl cyclase (sGC), which activates endothelial nitric oxide synthase (eNOS) to induce intracellular cGMP levels [10–13]. We confirmed that the PDE5 inhibitor, vardenafil (VDN), did not induce eNOS phosphorylation, whereas a single treatment of EGCG induced eNOS phosphorylation.

In this study, we showed that the inhibition of PDE5 potentiated EGCG-induced apoptosis in colorectal adenocarcinoma cells via the activation of the eNOS/cGMP/protein kinase Cδ (PKCδ) signaling pathway.

## 2. Materials and Methods

### 2.1. Cell Culture and Cell Viability

Cell density at $2 \times 10^4$ cells/mL of Caco-2 cells was cultured in DMED supplemented with 1% fetal bovine serum, at 37 °C in 5% $CO_2$ and 100% humidity. Cell viability was measured after treatment at 96 h through trypan-blue analysis in Caco-2 cells. EGCG was purchased from Sigma-Aldrich (St. louis, MO, USA). VDN was purchased from Toronto Research Chemicals (Toronto, ON, Canada). Bay 41-2272 was purchased from Enzo Life Sciences (Exeter, UK). Anti-PKCδ antibody was purchased from abcam. PDE1 inhibitor 8-Met-IBMX, PDE4 inhibitor rolipram, and PDE5 inhibitor VDN were purchased from Sigma-Aldrich. PDE2 inhibitor EHNA hydrochloride was purchased from abcam.

### 2.2. Analyzed Apoptotic Cell Death

Caco-2 cells ($2 \times 10^4$ cells/mL) were cultured in DMED supplemented with 1% fetal bovine serum, in 5% $CO_2$ and 100% humidity at 37 °C. Apoptotic cell death was determined using a flow cytometric test; the Annexin-V+ cells were evaluated by combining Annexin-regret V+/propidium iodide+ (early Annexin V+ propidium iodide-positive) and Annexin-V+/propidium iodide+ (late Annexin V+ propidium iodide- positive) cells after treatment with EGCG, Bay 41-2272, and/or VDN at 96 h analyses on a VerseTM system from BD.

### 2.3. The siRNAs Targeting PDE5

The siRNAs targeting PDE5 were purchased from Qiagen. Target sequences were as follows: siPDE5-1, 5′-CCAGCTTTACTGCCATTCAAT-3′; siPDE5-2, 5′-GCCATCTGCTTGCA-ACTGTAT-3′. Scrambled control siRNA or PDE5 siRNA and LipofectamineTM RNAiMAX Transfection Reagent were used for RNAi transfections (Life Technology, Carlsbad, CA, USA), which were performed according to the manufacturer's instructions, as previously described [13].

### 2.4. Western Blotting Analysis

Caco-2 cells were seeded at $1 \times 10^6$ cells/mL or $2 \times 10^4$ cells/mL and treated with 5 μM EGGC and/or 5 μM VDN for 3 h or 96 h for each experiment. Cells were lysed in lysis buffer (50 mM Tris-HCl (pH 7.5), 30 mM $Na_4P_2O_7$, 1 mM pervanadate, 150 mM NaCl, 1% Triton X-100, 50 mM NaF, 1 mM ethylenediaminetetraacetic acid, 2 mg/mL aprotinin, and 1 mM phenylmethanesulfonyl fluoride). Protein (approximately 50 μg) was suspended in Laemmli sample buffer (0.1 M Tris-HCl buffer, pH 6.8; 0.05% mercaptoethanol; 1% SDS; 0.001% bromophenol blue; and 10% glycerol). After boiling, the sample was electrophoresed on SDS-polyacrylamide gels. Gels were then electroblotted onto Trans-Blot nitrocellulose membranes (Bio-Rad, Berkeley, CA, USA). Incubation with the indicated antibodies (primary antibody) was completed in Tween 20-PBS containing BSA (1%). Blots were washed with Tween 20-PBS and incubated in antirabbit HRP conjugates (secondary antibody). After washing, indicated proteins were detected using an enhanced chemiluminescence system according to the manual from Amersham Life Sciences [13]. Samples were incubated overnight at 4 °C with the primary antibody that was used at 1:1000 dilution. Secondary antibody at 1:10000 dilution was incubated for 1 h. p-PKCδ at Ser664 antibody was obtained from Thermofisher. Anti-eNOS antibody was obtained from abcam. P-eNOS at Ser1177 antibody was obtained from BD Biosciences (San Jose, CA,

USA). Cleaved caspase-3 antibody (Asp175) was obtained from Cell Signaling Technology (Danvers, MA, USA). Anti-β actin antibody was purchased from Sigma-Aldrich, St. Louis, MO, USA. The siRNAs targeting PDE5 were purchased from Qiagen.

### 2.5. Statistical Analysis

Our data are indicated mean ± SEM compared with the controls. The IC50 values and isobologram methods were determined by using GraphPad Prism 5 software. The significant differences were assessed using Tukey's test. Statistical analyses were assessed using KyPlot 6.0 software (Kyens Lab, Tokyo, Japan). The level of synergistic effects was assessed by isobologram analyses using Graphpad prim 5 software (Dotmatics, Boston, MA, USA).

## 3. Results

### 3.1. Combination of EGCG and VDN Synergistically Induces Cell Death in Colorectal Adenocarcinoma

EGCG induces an anticancer effect by targeting the cell surface protein, 67LR [2]. However, the physiological concentration of EGCG is limited [4]. We evaluated the viability of Caco-2 cells treated with EGCG and VDN. We showed that EGCG dose-dependently suppressed the growth of Caco-2 cells with IC50 of 21.7 μM (Figure 1A). Our data also showed that the PDE5 inhibitor dose-dependently inhibited the growth of Caco-2 cells with an $IC_{50}$ of 31.6 μM (Figure 1B). Moreover, VDN potentiated the cell death effect of EGCG, with 50% inhibitory concentration ($IC_{50}$) values of 11.8 or 5.7 μM (Figure 1C,D). The results of the isobologram method indicated that the combination of EGCG and VDN synergistically induced cell death in colon cancer cells, including Caco-2 cells (Figure 1E) and HCT116 cells (Figure 1F).

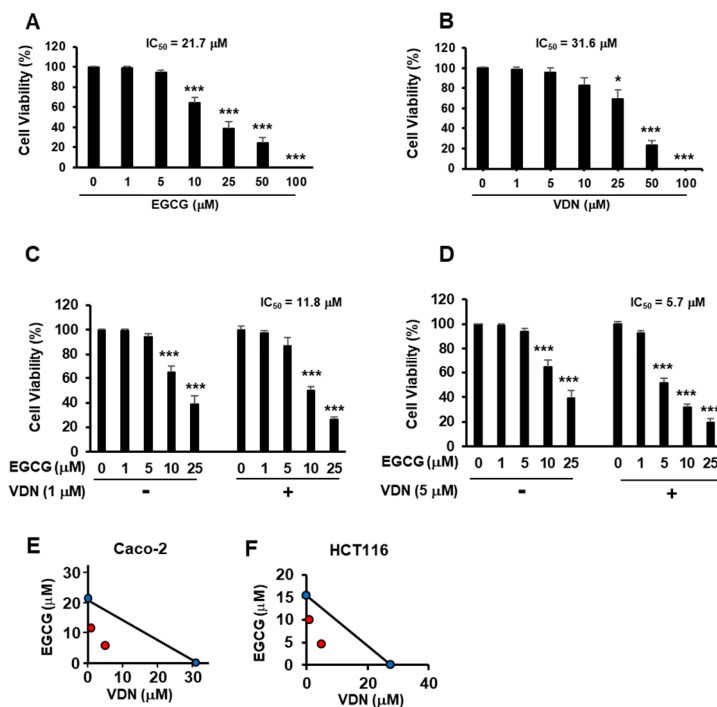

**Figure 1.** Combination of EGCG and VDN synergistically induces cell death in colorectal adenocarcinoma. (**A–D**) Viability of Caco-2 cells cultured with EGCG and the PDE5 inhibitor vardenafil (VDN) for 96 h. (**E,F**) Synergistic effects of EGCG and VDN measured using the isobologram analyses. Data are presented as mean ± SEM (*n* = 3); * *p* < 0.05, *** *p* < 0.001.

### 3.2. Combination of EGCG and VDN Induces Apoptosis in Colorectal Adenocarcinoma

To evaluate the combination of EGCG and VDN that induces apoptosis in Caco-2 cells, the cells were treated with 5 μM EGCG and 5 μM VDN. We found that the combination of EGCG and VDN induced apoptotic cell death in approximately 61.75% of cells, while a single treatment of EGCG or VDN did not affect the apoptosis levels relative to those of the control group (7.2%) (Figure 2A). Moreover, the combination of EGCG and VDN increased the levels of cleaved caspase-3, a crucial mediator of apoptosis (Figure 2B).

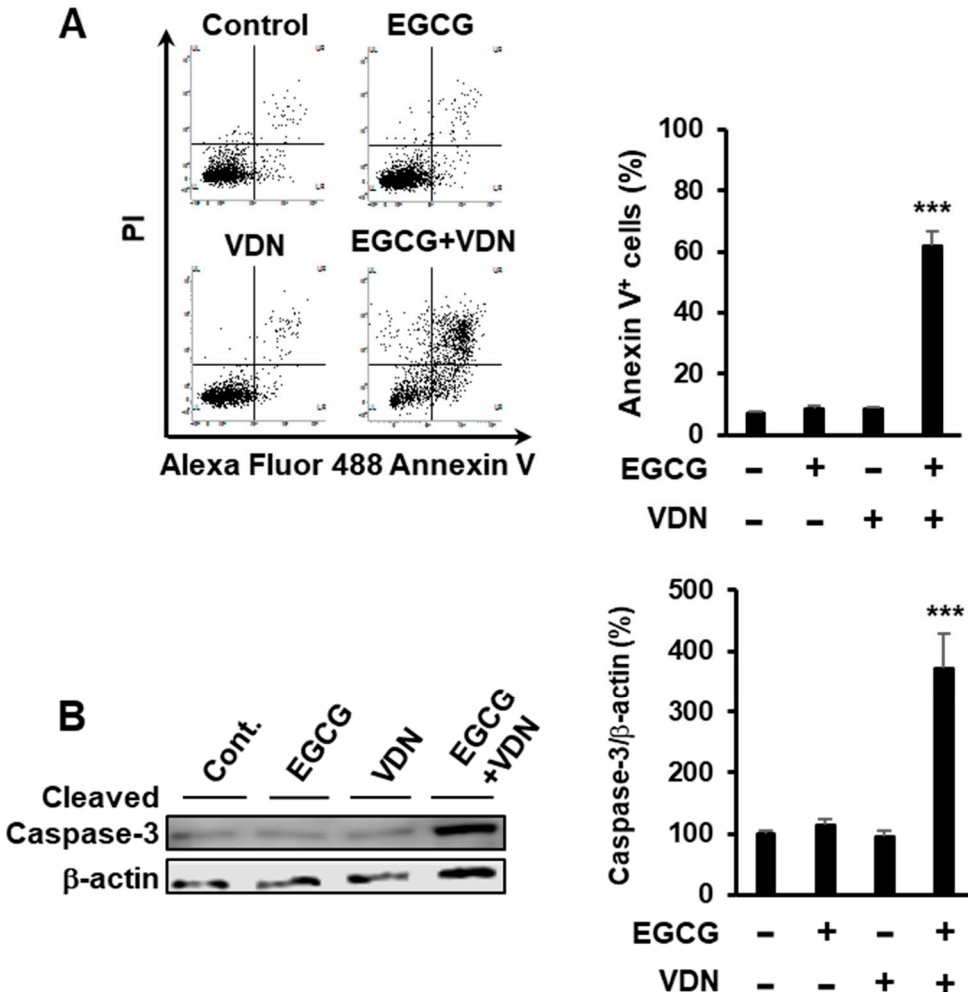

**Figure 2.** Combination of EGCG and VDN induces apoptosis in colorectal adenocarcinoma. (**A**) Caco-2 cells treated with 5 μM EGCG and 5 μM VDN for 96 h. Cells were stained with Annexin V–Alexa Fluor 488 and propidium iodide. Apoptotic cells were measured using flow cytometric analysis. Data are presented as mean ± SEM (*n* = 4); *** *p* < 0.001 (**B**) Cells were treated with 5 μM VDN and 5 μM EGCG was added for 96h. Levels of the apoptosis mediator cleaved caspase-3 were measured using Western blotting. Data are presented as mean ± SEM (*n* = 3); *** *p* < 0.001.

### 3.3. Expression of PDE5 Attenuates EGCG-Induced Apoptotic Cell Death in Colorectal Adenocarcinoma

We hypothesized that the PDEs protect colon cancer cells from EGCG-induced apoptotic cell death by downregulating the intracellular cGMP production. To determine the effect of various PDEs on the anti-colon-cancer effect of EGCG, Caco-2 cells were pretreated with PDE1 inhibitor 8-Met-IBMX (10 mM), PDE2 inhibitor EHNA hydrochloride (5 μM), PDE4 inhibitor rolipram (10 μM), or PDE5 inhibitor VDN (5 μM), then treated or not with EGCG (5 μM) for 96 h. Caco-2 cell death was induced by the combination of EGCG and selective PDE5 inhibitor VDN. This result suggested that PDE5 is a major negative regulator

of cGMP signaling (Figure 3A). To establish the role of PDE5 in the resistance of EGCG, we evaluated the effect of PDE5 knockdown in Caco-2 cells (Figure 3B left). We found that only PDE5 inhibitor VDN potentiated EGCG-induced cell death (Figure 3B right).

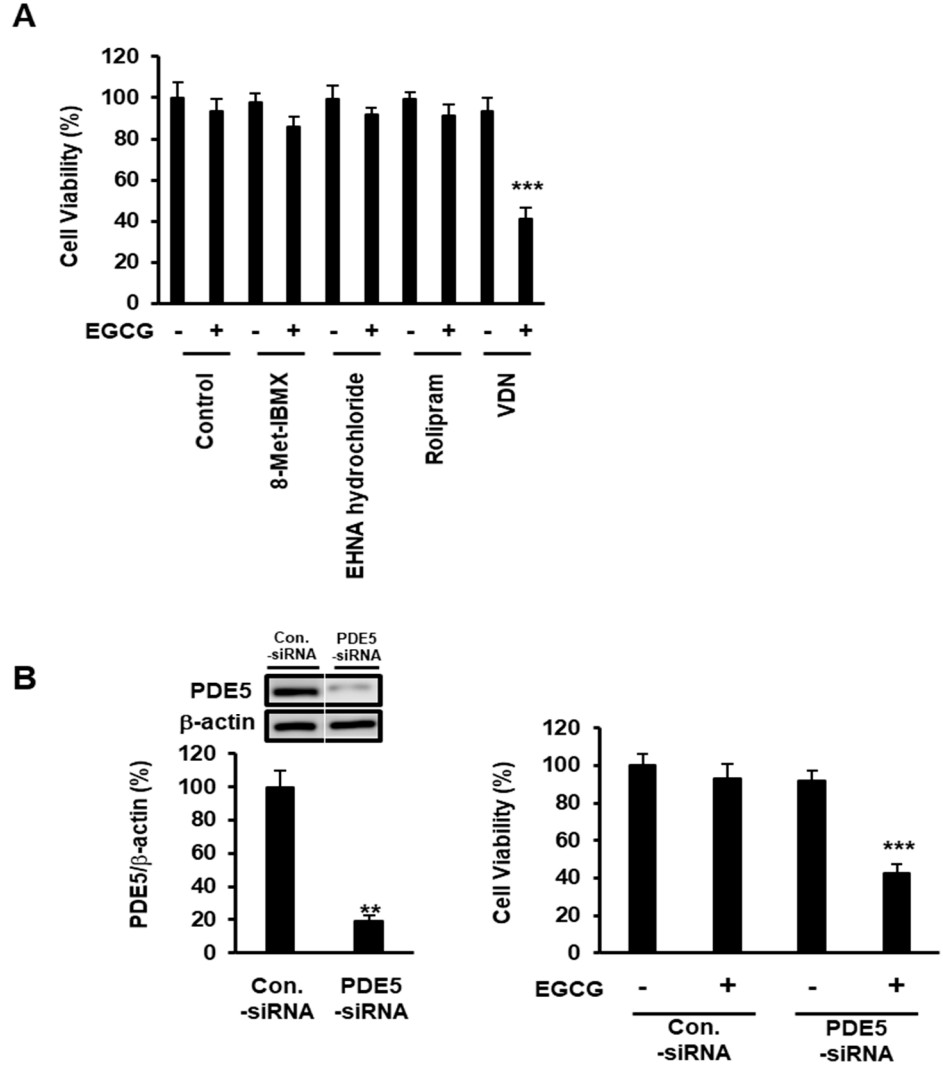

**Figure 3.** Expression of PDE5 attenuates EGCG-induced apoptotic cell death in colorectal adenocarcinoma. (**A**) Caco-2 cells were pretreated with PDE1 inhibitor 8-Met-IBMX (10 μM), PDE2 inhibitor EHNA hydrochloride (5 μM), PDE4 inhibitor rolipram (10 μM), or PDE5 inhibitor VDN (5 μM) for 96 h, then treated or not with EGCG (5 μM) for 96 h. (**B**) Left: Immunoblot analyses of PDE5 in Caco-2 cells. Right: Sensitivity of EGCG (5 μM for 96 h) of Caco-2 cells after knockdown of PDE5 expression. Data are presented as mean ± SEM ($n$ = 3); *** $p < 0.001$.

### 3.4. Activation of cGMP with Inhibition of PDE5 Activity Induces Cell Death in Colorectal Adenocarcinoma

To investigate whether the activation of cGMP signaling pathway induces cell death, Caco-2 cells were treated with Bay 41-2272 for the activation of intracellular cGMP production and VDN to inhibit PDE5 enzyme activity.

We found that the inhibition of PDE5 activity potentiated cGMP-mediated cell death, with the IC$_{50}$ values of Bay 41-2272 being 5.1 μM (combination treat with 1 μM VDN) or 3.2 μM (combination treat with 5 μM VDN), while the IC$_{50}$ for single treatment of Bay 41-2272 was 7.3 μM in Caco-2 cells (Figure 4A,B). The combination of Bay 41-2272 and VDN synergistically induced cell death (Figure 4C).

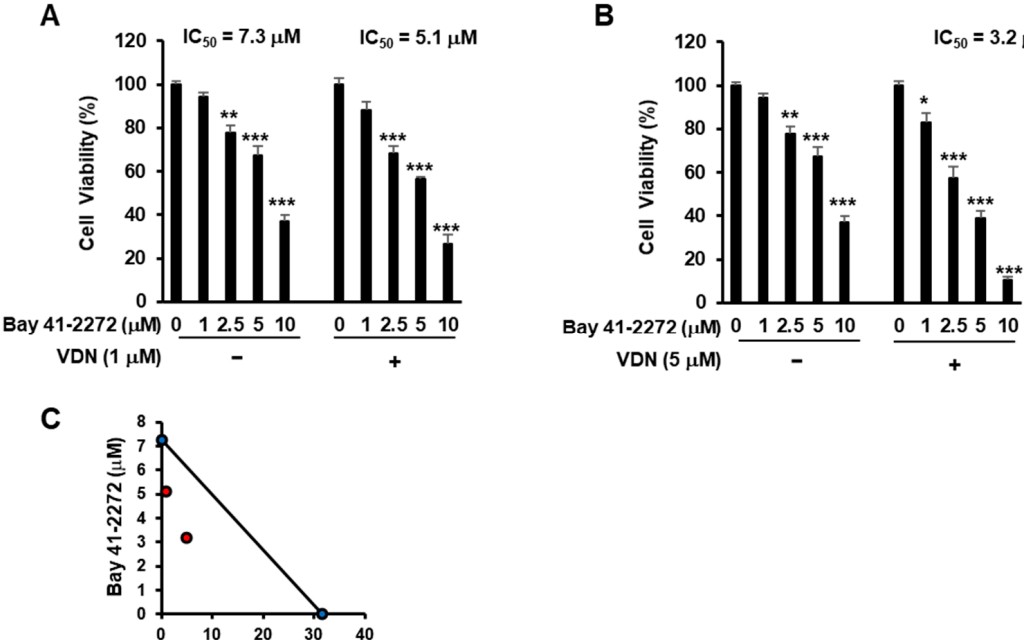

**Figure 4.** cGMP activation with inhibition of PDE5 activity induces cell death in colorectal adenocarcinoma. (**A**,**B**) Viability of Caco-2 cells cultured with the cGMP activator Bay 41-2272 and the PDE5 inhibitor VDN for 96 h. (**C**) Synergistic effects of Bay 41-2272 and VDN were measured using isobologram method. Data are presented as mean ± SEM ($n = 3$); * $p < 0.05$, ** $p < 0.01$, *** $p < 0.001$.

### 3.5. Inhibition of PDE5 Potentiates cGMP-Mediated Apoptosis in Colorectal Adenocarcinoma

We determined whether the combination of Bay 41-2272 and VDN induces apoptosis. We found that the combination of Bay 41-2272 and VDN induced apoptosis in 73.57% of cells, while a single treatment of VDN did not affect apoptosis levels in Caco-2 cells (Figure 5).

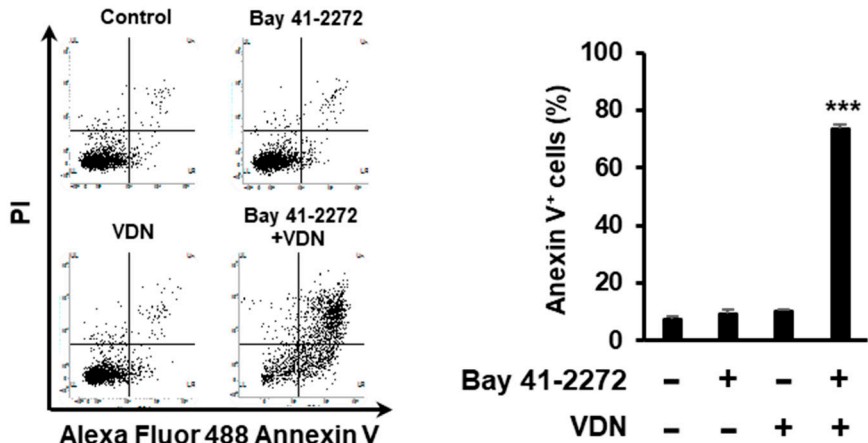

**Figure 5.** Inhibition of PDE5 potentiated cGMP-mediated apoptosis in colorectal adenocarcinoma. Caco-2 cells cultured with 5 µM Bay 41-2272 and 5 µM VDN for 96 h. Cells were stained with Annexin V–Alexa Fluor 488 and propidium iodide for flow cytometric analysis. Data are presented as mean ± SEM ($n = 4$); *** $p < 0.001$.

### 3.6. Inhibition of PDE5 Potentiates EGCG-Induced Apoptotic Cell Death via the eNOS/cGMP/PKCδ Signaling Pathway

To determine the effect of inhibition of PDE5 activity and EGCG on the up- and downstream signals of cGMP, we evaluated the effect of 5 µM VDN on EGCG-derived

eNOS and PKCδ phosphorylation. EGCG induced eNOS phosphorylation, but VDN did not affect the EGCG-induced eNOS phosphorylation at Ser1177 (Figure 6A). EGCG-induced cell death was inhibited by NS-2028, a specific sGC inhibitor (Figure 6B). Moreover, the combination of EGCG and VDN upregulated p-PKCδ phosphorylation at Ser664 (Figure 6C). Our findings indicated that the inhibition of PDE5 enzyme activity by VDN potentiated EGCG-induced cell death by increasing the physiological concentration of EGCG through the activation of the eNOS/cGMP/PKCδ signaling pathway.

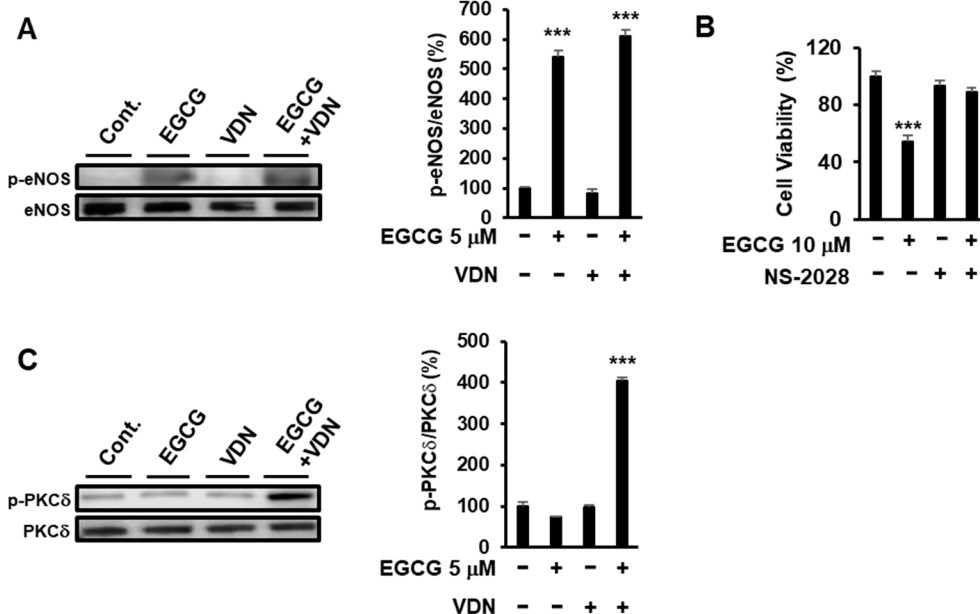

**Figure 6.** Inhibition of PDE5 potentiates EGCG-induced apoptotic cell death via the eNOS/cGMP/PKCδ signaling pathway. (**A**) Caco-2 cells were treated with 5 μM EGCG and 5 μM VDN for 1 h. Phosphorylation of eNOS at Ser1177 was measured using Western blotting. (**B**) Cells were treated with 10 μM EGCG and 5 μM of soluble guanylyl cyclase inhibitor (NS-2028). After 96 h, cell viability was determined. (**C**) Cells were treated with 5 μM EGCG and 5 μM VDN for 3 h. Phosphorylation of PKCδ at Ser664 was measured using Western blotting. Data are presented as mean ± SEM ($n = 3$); *** $p < 0.001$.

## 4. Discussion

Green tea has several health benefits including anticancer effects [14,15]. EGCG is a major component of green tea [1]. Previously, we showed that 67LR is a cancer-specific death receptor [2,3]. Moreover, we identified 67LR as a receptor for EGCG that is expressed in various types of cancers including human colorectal carcinoma Caco-2 cells [2,3,16,17]. In this pathway, cGMP initiates cancer-specific apoptotic cell death. Moreover, induced cGMP production is a rate-determining process of 67LR-dependent apoptotic cell death that activated by the EGCG, known as a natural ligand of 67LR [2,3]. However, the plasma concentration of EGCG is limited, and its anti-colon-cancer effect remains unelucidated. We showed that knockdown of PDE5 was sufficient to enhance the effect of EGCG. We demonstrated that the inhibition of PDE5 enzyme activity potentiated EGCG-induced apoptotic cell death with a physiological concentration of EGCG (5 μM) via the activation of the eNOS/cGMP/PKCδ signaling pathway in colon cancer cells.

PDE5 is an enzyme and family of PDEs that degrades the phosphodiester bond in the intracellular second messenger cGMP [5]. A clinical study showed that PDE5 inhibitors are well known for their health benefits in various types of diseases, including erectile dysfunction, heart failure, and pulmonary hypertension [18–21]. However, not only these PDE5 inhibitors may be effective in cancer treatment. A preclinical study reported that widely used PDE5 inhibitors are candidates as anticancer agents [22], and PDE5

inhibitors could be candidates for anticancer drugs because they are relatively safe [22]. Moreover, the upregulation of the cGMP signaling pathway plays a crucial role in anticancer effects [23–28].

PDE5 is overexpressed in various cancer cell types including colon cancer, stomach cancer, chronic lymphocytic leukemia, prostate cancer, acute myeloid leukemia, pancreatic cancer and breast cancer [3,17,18,21,23,24]. However, little is known about the role of PDE5 in colon cancer cells. We investigated whether PDE5 inhibition is sufficient to enhance the anticancer effect of PDE5. Caco-2 cells were knocked down of PDE5, and we assessed the sensitivity to EGCG. Our results directly showed that PDE5 knockdown was sufficient to enhance the effect of EGCG (Figure 3B). As vardenafil is a clinically used PDE5i and significantly increased the effect of EGCG (Figure 1), PDE5 is a candidate to enhance the anticancer effect of EGCG on colon cancer. Because each antibody has a different affinity, we could not compare the expression levels of different PDEs. Instead of those approaches, we assessed the effect of different PDEs on the effect of EGCG with PDE5i VDN. Our results indicated that VDN significantly enhanced the anticancer effect of EGCG. We also showed that a single treatment of 5 µM EGCG or 5 µM VDN did not show a cell death effect, while the combination of EGCG and VDN synergistically induced apoptotic cell death. We confirmed that EGCG induced eNOS phosphorylation at Ser1177, while a single treatment of VDN did not induce eNOS phosphorylation at Ser1177. These results suggested that the inhibition of PDE5 activity potentiates the cell death effect of EGCG through enhancing the downstream molecules of cGMP; nonetheless, the inhibition of PDE5 activity did not affect the upstream molecules of cGMP in colon cancer cells. The value of our findings is notable because several PDE5 inhibitors including VDN are approved by the FDA [29].

High doses of EGCG can induce hepatotoxicity [30]; in clinical trials, elevation of AST/ALT levels has been observed [4]. Importantly, the combination of EGCG and VDN did not increase the AST/ALT levels in the serum. From our perspective, PDE5 expression in cancer cells may one reason why higher concentrations of EGCG are needed to induce apoptosis in colon cancer cells, although the dissociation constant (Kd) of EGCG that binds to cell surface protein 67LR is only 0.04 µM [2]. Its poor bioavailability is one of the reasons why clinical use of EGCG is difficult.

PKCδ plays a crucial role in proapoptotic signaling in many types of cells [31]. For example, a synthetic sphingosine immunosuppressant, FTY720, promotes the phosphorylation of PKCδ at Tyr311, leading to the autophosphorylation of PKCδ at Ser664, which subsequently activates caspases [32]. We previously reported that PKCδ is essential for the anticancer effect of EGCG and EGCG-induced cell death [14]. Our study showed that VDN potentiated EGCG-induced phosphorylation of PKCδ at Ser664 and the upregulation of cleaved caspase-3 levels, indicating the proapoptotic signaling pathway in colon cancer cells.

Taken together, our findings show that the inhibition of PDE5 activity potentiates EGCG-induced apoptotic cell death in colon cancer cells, accompanied by the activation of the eNOS/cGMP/PKCδ signaling pathway. Moreover, in this study, we used the plasma concentration of EGCG, which is noteworthy and could be effective for chemotherapy in colorectal adenocarcinoma cells.

**Author Contributions:** J.B., K.L., H.T., Y.F., M.K., S.-J.L. and S.-J.P. contributed to the research idea and design. J.B., H.T., Y.F. and M.K. created the search strategy. J.-S.P., J.J., H.T., Y.F., M.K., J.S.L., Y.-C.C., S.-J.L. and S.-J.P. screened titles, abstracts, and full texts. J.B., K.L., J.-S.P., J.J. and S.-J.P. contributed to data extraction and quality assessment. J.B., K.L., J.-S.P., J.J. and M.K. contributed to the statistical analysis and interpretation of data. J.B. wrote the first draft of the manuscript. H.T., J.S.L., Y.-C.C., S.-J.L. and S.-J.P. edited the draft of the manuscript. All authors have read and agreed to the published version of the manuscript.

**Funding:** This work was supported by the National Research Foundation of Korea (NRF) grant funded by the Korea government (MSIT) (NRF-2021R1C1C2095006). This research was supported by the Korea Institute of Planning and Evaluation for Technology in Food, Agriculture, Forestry and Fisheries (iPET) through Animal Disease Management Technology Development Program (118097-2) funded by the Ministry of Agriculture, Food and Rural Affairs (MAFRA) and the KRIBB Research Initiative Program (KGM5242113), Republic of Korea.

**Institutional Review Board Statement:** Not applicable.

**Informed Consent Statement:** Not applicable.

**Data Availability Statement:** The data that support the findings of this study are available upon request from the corresponding author.

**Conflicts of Interest:** The authors declare no conflict of interest.

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
