# Peer review of "Phosphodiesterase 5 Inhibitor Potentiates Epigallocatechin 3-O-Gallate-Induced Apoptotic Cell Death via Activation of the cGMP Signaling Pathway in Caco-2 Cells"

_cimb, doi:10.3390/cimb44120426_

Round 1
Reviewer 1 Report
This manuscript demonstrates that combination of EGCG and a PDE5 inhibitor (VDN) synergistically induced cell death and apoptosis in Caco-2 cells. Moreover, the authors show that PD5E expression attenuates EGCG-induced apoptotic cell death in Caco-2 cells. Furthermore, the authors show that eNOS/cGMP/PKC signaling pathway is involved in EGCG-induced apoptotic cell death in Caco-2 cells. This reviewer thinks the content of this manuscript is interesting. The manuscript could be improved if note is taken of the following points:
1. The authors should explain only cell culture in the section on cell culture in Materials and Methods.
2. The authors should also explain in a separate section how cell viability was examined, how apoptosis was examined, and information on purchased materials.
3. The authors should explain in detail about western blot analysis. The authors should at least explain protein quantification, antibody concentration, and reaction time. The authors should also describe densitometry analysis in detail.
4. The authors should elaborate on the isobologram method.
5. The authors should explain how they obtained 8-Met-IBMX, EHNA hydrochloride, and roliparam in Materials and Methods.
6. The picture of cleaved caspase-3 in Fig.2 and the pictures of eNOS and PKC in Fig.6 are too dark. The authors should use better pictures of western blot analysis in the manuscript.
7. This reviewer thinks that the authors should investigate whether caco-2 expresses 67LR.
Author Response
This manuscript demonstrates that combination of EGCG and a PDE5 inhibitor (VDN) synergistically induced cell death and apoptosis in Caco-2 cells. Moreover, the authors show that PD5E expression attenuates EGCG-induced apoptotic cell death in Caco-2 cells. Furthermore, the authors show that eNOS/cGMP/PKC signaling pathway is involved in EGCG-induced apoptotic cell death in Caco-2 cells. This reviewer thinks the content of this manuscript is interesting. The manuscript could be improved if note is taken of the following points:
We thank the reviewer for acknowledging the importance of our paper. We were delighted to read the positive comments and are grateful for the helpful suggestions. As suggested by the reviewer, we added the reference about the Caco-2 expresses 67LR and of their potential application and related description. We corrected the manuscript and amended the expression to clarify.
Point 1: The authors should explain only cell culture in the section on cell culture in Materials and Methods.
Response 1: Thank you for your suggestion. Accordign to your comments, we are corrected in the section on cell culture and cell viability in Materials and Methods.
We added the description in Materials and Methods (2.1. Cell culture and cell viability) section at page 2, line 16 as described below.
Point 2: The authors should also explain in a separate section how cell viability was examined, how apoptosis was examined, and information on purchased materials.
Response 2: Thank you for your helpful advice. As suggested by the reviewer, We have corrected explain in a separate section how cell culture and cell viability was examined, how analyzed apoptotic cell death was examined, and information on purchased materials in Materials and Methods (Please, show the page 2).
Point 3: The authors should explain in detail about western blot analysis. The authors should at least explain protein quantification, antibody concentration, and reaction time. The authors should also describe densitometry analysis in detail.
Response 3: Thank you for your comments. As suggested by the reviewer, We have corrected at least explain protein quantification, antibody concentration, and reaction time. We also describe densitometry analysis in detail in Materials and Methods.
“Protein (approximately 50 μg) was suspended in Laemmli sample buffer (0.1 M Tris-HCl buffer, pH 6.8; 0.05% mercaptoethanol; 1% SDS; 0.001% bromophenol blue; and 10% glyc-erol), After boiled electrophoresed on SDS-polyacrylamide gels. Gels were then electrob-lotted onto Trans-Blot nitrocellulose membranes (Bio-Rad). Incubation with the indicated antibodies (primary antibody) was done in Tween 20-PBS containing BSA (1%). Blots were washed with Tween 20-PBS and incubated in anti-rabbit HRP conjugates (secondary antibody) . After washing, indicated proteins were detected using an enhanced chemilu-minescence system according to the manual from Amersham Life Sciences. [13]. Samples were incubated overnight at 4°C with the primary antibody that were used at 1:1000 dilu-tion. Secondary antibody at 1:10000 dilution and incubated for 1 hour.”
We added the description in 2.5. Western blotting analysis section (Please, show the page 2-3).
Point 4: The authors should elaborate on the isobologram method.
Response 4: Thank you for your suggestion. We have corrected elaborate on the isobologram method in Materials and Methods.
“The level of Synergistic effects was assessed by isobologram analyses using Graphpad prim5 software from Dotmatics.”
We added the description in Materials and Methods section (2.6. Statistical analysis) at page 3, line 18-19 described below.
Point 5: The authors should explain how they obtained 8-Met-IBMX, EHNA hydrochloride, and roliparam in Materials and Methods.
Response 5: Thank you for your helpful advice. As suggested by the reviewer, We have corrected explain how we obtained 8-Met-IBMX, EHNA hydrochloride, and roliparam in Materials and Methods.
“PDE1 inhibitor 8-Met-IBMX, PDE4 inhibitor rolipram and PDE5 inhibitor VDN was pur-chased from Sigma-Aldrich. PDE2 inhibitor EHNA hydrochloride was purchased from abcam.”
We added the description in Materials and Methods section (2.1. Cell culture and cell viability) at page 2 , line 19-24 as described below.
Point 6: The picture of cleaved caspase-3 in Fig.2 and the pictures of eNOS and PKC in Fig.6 are too dark. The authors should use better pictures of western blot analysis in the manuscript.
Response 6: We have corrected the manuscript according to the comments. We have corrected the better pictures of cleaved caspase-3 in Fig.2 and the pictures of eNOS and PKC in Fig.6.
Point 7: This reviewer thinks that the authors should investigate whether caco-2 expresses 67LR.
Response 7: Thank you for your comments. Sorry for the confusion and our explanation is insufficient. Previously, we reported that human colon adenocarcinoma Caco-2 cells exhibited higher expression level of 67LR. As suggested by the reviewer, we added the references about the caco-2 expresses 67LR.
“Previously, we showed that 67LR is a cancer-specific death receptor [2,3]. Moreover, we identified the 67LR is a receptor for EGCG that expressed in various types of cancers in-cluding human colorectal carcinoma Caco-2 cells [2,3,16,17].”
We added the description in discussion section at page 8, line 20-23 as described below.

Reviewer 2 Report
The manuscript described a study on the synergistic effect of Phosphodiesterase 5 inhibitor (PDE5) and EGCG on the colon cancer in vitro. The reviewer recommends the publication of the manuscript after the authors the following concerns
1) PDE5 inhibitors are used in the treatment of ED, pulmonary hypertension, which is a class of drugs with cardiovascular benefits. But have this class of drugs been used for treatment of cancers before in clinical application, the authors should provide background information
2) As the author stated in discussion "the plasma concentration of EGCG is limited, and its anti-colon cancer effect remains unelucidated“,but "VDN potentiated cell death effect of EGCG, with 50% inhibitory concentration (IC50) values of 11.8 or 5.7 μM (Figure 1 C, D)." , Are these concentrations too high for in vivo applications?
3) the authors mentions cell surface protein 67-kDa laminin receptor (67LR) for 37 EGCG, but is this information relevant to the study in the manuscript?
4) Figure 4A is not necessary
Author Response
Response to Reviewer 2 Comments
The manuscript described a study on the synergistic effect of Phosphodiesterase 5 inhibitor (PDE5) and EGCG on the colon cancer in vitro. The reviewer recommends the publication of the manuscript after the authors the following concerns
Point 1. PDE5 inhibitors are used in the treatment of ED, pulmonary hypertension, which is a class of drugs with cardiovascular benefits. But have this class of drugs been used for treatment of cancers before in clinical application, the authors should provide background information
Response 1: Thank you for your suggestion. As suggested by the reviewer, We have corrected the explain the background information.
A clinical study showed that PDE5 inhibitors are well known for their health benefits in various types of diseases, including erectile dysfunction, heart failure, and pulmonary hypertension [18-21].
“However, not only these effect PDE5 inhibitors are may be effective in cancer treatments. A pre-clinical, has been presented to make the case that widely used PDE5 inhibitors are candidates for anticancer agents [22] and PDE5 inhibitors could be potential candidates for anti-cancer drug because those are relatively safe.”
We added the description in Discussion section at page 9, line 1-5 as described below.
Point 2. As the author stated in discussion "the plasma concentration of EGCG is limited, and its anti-colon cancer effect remains unelucidated“,but "VDN potentiated cell death effect of EGCG, with 50% inhibitory concentration (IC50) values of 11.8 or 5.7 μM (Figure 1 C, D).", Are these concentrations too high for in vivo applications?
Response 2: Thank you for your comments. Previouse clinical study demonstrated the plasma concentration of EGCG is about under 10 uM [4]. Considering that IC50 of EGCG with PDE5 inhibitor is 5.7 mM. Those concentration cloud be achievable in clinical setting.
Reference
[4] Shanafelt, T.D. et al. Phase I trial of daily oral polyphenon E in patients with asymptomatic rai stage 0 to II chronic lymphocytic leukemia. J. Clin. Oncol. 2009, 27, 3808-3814.
Point 3. the authors mentions cell surface protein 67-kDa laminin receptor (67LR) for EGCG, but is this information relevant to the study in the manuscript?
Response 3: Thank you for your comments. To explain our background, we mentioned a cancer-specific death receptor 67LR for EGCG
“Previously, we showed that 67LR is a cancer-specific death receptor [2,3]. Moreover, we identified the 67LR is a receptor for EGCG that expressed in various types of cancers in-cluding human colorectal carcinoma Caco-2 cells [2,3,16,17]. In this pathway, cGMP initi-ated cancer-specific apoptotic cell death. Moreover, induced cGMP production was a rate-determining process of 67LR-dependent apoptotic cell death that activated by the EGCG, known as a natural ligand of 67LR [2,3].”
We added the description in Discussion section at page 8, line 20-25 as described below.
Point 4. Figure 4A is not necessary
Response 4: Accordign to your commnets, we have corrected the Figure 4.
